A low-cost unity-based virtual training simulator for laparoscopic partial nephrectomy using HTC Vive

Rasheed Fareeha 1
Bukhari Faisal 1 faisal.bukhari@pucit.edu.pk
Iqbal Waheed 1
http://orcid.org/0000-0003-1839-2527 Asif Muhammad 2
Chaudhry Hafiza Ayesha Hoor 3
1 Department of Data Science, University of the Punjab , Lahore , Pakistan
2 Department of Computer Science, National Textile University , Faisalabad , Pakistan
3 Department of Computer Science, University of Turin , Piedmont , Italy
Balas Valentina Emilia
Electronic publication date: 2023 Oct 17
Publication date: 2023
Volume: 9
Electronic Location ID: e1627
Received 2023 Jun 27; Accepted 2023 Sep 11
Copyright: © 2023 Rasheed et al.
Copyright year: 2023
Copyright holder: Rasheed et al.
License: This is an open access article distributed under the terms of the Creative Commons Attribution License, which permits unrestricted use, distribution, reproduction and adaptation in any medium and for any purpose provided that it is properly attributed. For attribution, the original author(s), title, publication source (PeerJ Computer Science) and either DOI or URL of the article must be cited.
License URL: https://creativecommons.org/licenses/by/4.0/

Keywords: Simulation, Computer vision, Virtual training

Funding: The authors received no funding for this work.

==============================
Laparoscopic education and surgery assessments increase the success rates and lower the risks during actual surgeries. Hospital residents need a secure setting, and trainees require a safe and controlled environment with cost-effective resources where they may hone their laparoscopic abilities. Thus, we have modeled and developed a surgical simulator to provide the initial training in Laparoscopic Partial Nephrectomy (LPN—a procedure to treat kidney cancer or renal masses). To achieve this, we created a virtual simulator using an open-source game engine that can be used with a commercially available, reasonably priced virtual reality (VR) device providing visual and haptic feedback. In this study, the proposed simulator’s design is presented, costs are contrasted, and the simulator’s performance is assessed using face and content validity measures. CPU- and GPU-based computers can run the novel simulation with a soft body deformation based on simplex meshes. With a reasonable trade-off between price and performance, the HTC Vive’s controlled soft body effect, physics-based deformation, and haptic rendering offer the advantages of an excellent surgical simulator. The trials show that the medical volunteers who performed the initial LPN procedures for newbie surgeons received positive feedback.

Introduction

Although males have twice as high a risk of having renal cancer as females, it is one of the top ten tumors in both sexes (Siegel et al., 2022). Since the 1960s, when open radical nephrectomy (ORN) was first developed, the mainstay of treatment for restricted renal cell carcinoma (RCC) has been the surgical excision of kidney tumors by ORN (Robson, 1963; Robson, Churchill & Anderson, 1969). Laparoscopic radical nephrectomy (LRN) became the new acknowledged standard approach for a surgical recession of renal cancer following the first report of a small renal tumor treated with LRN was reported in 1991 (Clayman et al., 1991). A small incision is made on the patient’s body during a laparoscopic nephrectomy, and a laparoscopic instrument is then slid into the body cavity. A tiny camera is used to help the surgeon see the nearby organs as they operate within the bodily cavity to remove the renal tumor. Compared to open radical surgery, laparoscopic nephrectomy (LPN) is safer, recovers more quickly (because of the small incision), and yields superior results. Compared to open radical nephrectomy, laparoscopic nephrectomy is more challenging surgically and has a steep learning curve. It also demands excellent hand-eye coordination, extensive surgical practice, and technical training (Huang et al., 2015; Kapoor, 2009). Owing to these reasons, laparoscopic nephrectomy is still harder to adapt as the standard practice in developing countries.

Virtual reality is utilized to find solutions for these issues. Virtual reality not only enables distance learning and training in the medical field but also provides a suitable alternative to in-vivo surgeries (Chaudhry et al., 2020). The virtual realistic laparoscopic simulator enables surgeons to rehearse various laparoscopic techniques, including clipping, cutting, suturing, knot tying, and camera handling. Because these laparoscopic procedures employ cutting-edge imaging and surgical methods, the post-surgery outcomes include shorter hospital stays and quicker recovery times. Additionally, it guarantees a less painful procedure with minimal blood loss and better cosmetic outcomes (Alaker, Wynn & Arulampalam, 2016; Mazurek et al., 2019). Simulators for virtual reality come in many different varieties today. The ultimate objective of most simulators is to give training methods that can help surgeons be ready for difficult surgeries in advance (Yiannakopoulou et al., 2015; Gupta, Cecil & Pirela-Cruz, 2020). Laparoscopic training, skill evaluation, and haptic feedback have all been extensively studied (Borglund et al., 2021; Lee & Lee, 2018; Chaudhry et al., 2020). Low-cost hardware systems with trainees’ evaluation still require improvement, especially in emerging and underdeveloped nations. These systems have been demonstrated to be an investment in improving surgeons’ surgical abilities through game-based training (Tanjung et al., 2020; Haowen et al., 2021).

Additionally, it tackles the ethical dilemma of practicing on other living things while increasing the trainees’ self-confidence, skills, and experience in a monitored and repeated environment (Soares et al., 2021; Zhang et al., 2018; Fürst et al., 2014). According to certain research (Craighead, Burke & Murphy, 2008; Ahamed et al., 2020), using Unity for real-time applications and simulation purposes has significant advantages. Due to its high location tracking precision and accuracy level, the HTC Vive VR device is drawing interest as a low-cost simulation tool. It is considered suitable for studies that do not require self-motion, thus proving it a good option for surgery training (Borges et al., 2018; Qiu, Jiang & Luo, 2022).

This study demonstrates a low-cost virtual training simulator for laparoscopic partial nephrectomy built on Unity and running on an HTC Vive. The primary takeaways from our research are: 1) Creation of an anatomical model for the kidney with renal cell carcinoma (RCC) and minimal mesh representation.

2) Proposing position based dynamics (PBD) with the force-based approach as a stable and effective solution for soft body deformation.

3) Creating a unique LPN training experience for surgeons using a game engine.

4) Exploring the potential applications of a low-cost HTC Vive VR system for training and haptic feedback.

5) Establishing the simulator’s validity metrics and a criterion-based system for surgeon evaluation.

Our suggested training method based on Unity follows a more straightforward deformation representation with less computing time than existing high-fidelity pricey trainers. About the earlier approaches, the proposed algorithm’s flexibility in adjusting the parameters gives users much control over the simulation. Using the HTC Vive HMD and controllers, the user can easily see and interact with the deformed body.

Details and challenges in LPN

LPN has emerged as a viable alternative to open nephrectomy while minimizing patient morbidity. The primary emphasis of this study is the treatment of exophytic renal tumors, which have benefited from improvements in laparoscopic experience. The Da Vinci Surgical System is usually used for surgical practices as it is safe and accurate. However, the machine is very costly. The main difficulties during the operating process that require practice are: 1) To prevent unnecessary harm to the kidney and its adjacent areas, the tumor’s precise location must be determined.

2) To stop unmanageable bleeding, the renal artery, renal vein, and ureter are clipped and mobilization of descending colon is ensured.

3) Adhesions between the kidneys can be successfully separated to avoid operation failure.

4) Laparoscopic tool insertion needs to be done cautiously since inexperienced handling can harm the intestines and major blood arteries.

The surgeon can practice and complete a variety of partial nephrectomies efficiently if they can manage this broad range of indications. Figure 1 is added here to show the steps involved in a typical nephrectomy procedure. The patient must be placed correctly, and the camera and other equipment must be inserted through trocars as the initial step. Figures 1A and 1B shows the mobilization of descending colon; it is necessary to forcibly ligate either the inferior mesenteric artery or the left colic artery in order to mobilise the descending colon. As a result, the inferior mesenteric artery stops contributing to the colon’s blood supply. The next step involves the identification of renal artery, veins and other features. Figures 1C and 1D represents clipping of renal artery and cutting the renal vein respectively. Then the final step is to dissect cancerous tissues around the kidney as shown in Fig. 1E. Our team has reproduced some of the same processes using the HTC Vive, which is significantly less expensive than the Da Vinci machines, by simulating some of the same phases using a simple mesh representation. We have focused on the important step of accurately separating cancerous tissues from the rest of the structure because the others can be practised in various training sessions. The surgeons starting their first LPN practise can use our solution.

Figure 1 Steps to identify the structure and perform laparoscopic nephrectomy using LAPVision.

Reprinted from Miyata et al. (2020).

Related work

To modernize the diagnosis and care of kidney cancer, Yu, Xu & Liu (2021) examined the literature on partial nephrectomy. The main benefit of this treatment is that it prevents total kidney loss; nonetheless, there is a chance of local infection and other problems. To treat autosomal dominant polycystic kidney disease (ADPKD), Verhoest et al. (2012) have reviewed some advantages of laparoscopic nephrectomy over open nephrectomy. They discovered that two patients who underwent open cyst ectomy experienced incisional hernias. Contrarily, with laparoscopic nephrectomy, there was minimal blood loss, less postoperative pain, and a quick recovery. Diaz et al. (2010) have covered some of the algorithms and approaches for surgical modeling of tissues and blood vessels for these models to be used effectively during training. Another exciting work by Qian et al. (2017) concentrated on simulation frameworks and effective computation methods rather than specific problems with deformation, force direction, and feedback. For realistic rendering, the techniques presented contain a wide range of intuitive parameters.

The customized finite element method (FEM) and extended finite element method (XFEM) algorithms used in this research (Kugelstadt, Koschier & Bender, 2018) provide ways to increase simulation efficiency and design procedure accuracy. For mapping nodal connectivity, the FEM approach typically needs a longer execution time and a more extensive data set. Due to nonlinear constraint handling and a fast convergence rate, the position-based dynamics (PBD) method is more practical and stable (Romeo, Monteagudo & Sánchez-Quirós, 2020; Bender et al., 2014; Khan, Choi & Hong, 2022). In contrast to the FEM method, it immediately modifies the vertex locations, skipping the velocity layer. Extended position-based dynamics (XPBD), which implicitly offers force estimates and makes simulations appear realistic visually but necessitates substantial computing power, is another helpful technique (Macklin, Müller & Chentanez, 2016).

The three methods for soft tissue deformation—linear elasticity theory, tensor-mass model, and spring-mass model have been highlighted by Witt, Schumann & Klimant (2019). All models’ computational complexity varies depending on how many model edges are involved. They created a novel hybrid technology that combines the advantages of earlier techniques for tissue cutting and tear modeling. The SOFA architecture is frequently used in cutting-edge works for simulations. It combines hyperelasticity with accurate object physics, which requires more processing. It functions nicely with haptic devices such as Geomagic, ARTTrack, Novint, Falcon, etc. (SOFA—Features, 2023, https://www.sofa-framework.org/about/features/). A change in the development process is necessary to keep up with the improvements and expanding requirements in the simulation of virtual healthcare environments. The method’s complexity and trainer costs must decrease. Low-cost virtual equipment is implemented to improve surgical skills, promoting market rivalry. Zhong et al. (2015) and Wheeler et al. (2018) have created plausible virtual settings using Unity 3D to find a cost-effective solution and restrict the issues associated with the regulated training environment. The game engines aid in quick prototyping and data gathering. Different commercially available head-mounted displays are employed nowadays in particular situations, and their performance is assessed depending on the functionality they offer (Tseng, 2016; Srivastava, Kumar & Zareapoor, 2018).

When compared to traditional approaches, this study (Vamadevan et al., 2023) discusses the employment of haptic and non-haptic responsive devices as the most efficient way to obtain surgical training with a sense of realism. The functioning technology, response to external forces, degree of freedom, authenticity, and immersive experience of each simulator were compared in their study. In their work, Nemani et al. (2018) evaluated the Fundamentals of Laparoscopic Surgery (FLS) and the Virtual Basic Laparoscopic Skill Trainer (VBLaST) trainers of a pattern cutting simulator using task performance scores. There is a sizable gap to be closed for advancements in accurate functioning, effective organ modeling, reduced computational time, and standardization of metrics used for surgeons’ evaluation. The continual assessment of current systems and the design of specific practice tasks, scores, and training duration as instructed by these works (Mao et al., 2021; Sommer et al., 2021) are crucial. The main topics of this study are the practical approach of soft body modeling, position-based mesh deformation, performance evaluation in terms of stability and speedy solving time, and haptic feedback from the system employing a less expensive VR device. Similarly, we have investigated the capabilities and constraints of this game engine by recreating the virtual environment through Unity, concentrating on the training purpose.

Modular simulation procedure views

An interactive user interface, a physical component for practicing the method, a renderer component for visualizing 3D elements, and an event management module make up the majority of training simulators. The same structure is the foundation of our simulator.

Figure 2 depicts the training environment’s modular views. In order to identify the patient’s malignant structure and other important aspects, the trainees must first need a CT scan result, as shown in Fig. 2A, which depicts the tumour on the left kidney. We prototyped the kidney model using a minor renal cell carcinoma case with an almost 3–5 cm peripheral mass, using subfigure a as a reference image for our soft body design. The next stage is to identify cancerous area and other important characteristics of the kidney model that was employed. Trainees can quickly mark and precisely cut along the malignant structure with the aid of two laser-emitting controllers while minimizing harm to the nearby tissues. The Unity profiler is used to capture the task’s performance time. The malignant portion of the kidney must be removed as the last stage.

Figure 2 Modular views of a simulated laparoscopic partial nephrectomy.

(A) Abdominal CT scan result. Reprinted from (Gharaibeh et al., 2022). (B) Anatomical structure design and labeling. (C) Laser cut across the cancerous part with profiler view. (D) Teasing away the cancerous part.

System design and physical simulation

The Unity physics engine is used to understand the physics behavior of the kidney organs’ soft body, which was created using Blender (https://www.blender.org/). The triangular meshes are cut with a specific force and in a precise direction throughout the rendering process to show the object’s physical appearance. An abstract representation of the entire system architecture is shown in Fig. 3.

Figure 3 Laparoscopic partial nephrectomy system architecture.

3D modeling

With the help of simplex meshes, which are topologically dual triangulations, we have modeled the kidney and the cancerous part. Depending on the local energy, these meshes are more adaptable. It is common practice to employ this method for effective physics-based modeling. It enables reliable and effective object deformation.

Joining components: Figure 4 shows the kidney model consisting of different 3D components (renal artery, vein, and cancerous part) extruded and then joined together using Blender’s Join operation. Figures 4A and 4B present the 3D structure of a healthy kidney with a solid and wireframe view, respectively. Pertinent materials have been used to understand better and match the organ’s color palette. Figure 4 shows a solid view in Fig. 4C and a wireframe view in Fig. 4D after the malignant area has been incorporated onto the mesh surface.

Figure 4 Step-wise 3D modeling of kidney organ and cancerous structure.

(A) 3D healthy kidney model solid view. (B) 3D healthy kidney model wireframe view. (C) 3D kidney model solid view. (D) 3D kidney model wireframe view. (E) 3D kidney model solid view after subdivision modifier. (F) 3D kidney model wireframe view after subdivision modifier.

Subdivision modifier: Subdivision surface modification divides the mesh triangles into subsurfaces. Only a few properties were changed; the mesh’s core topology remained unchanged. The models are depicted in Figs. 4E and 4F after the modifier has been applied. Since the mesh triangles are clearly sparse, the rocky mesh surface in Fig. 4C is not plausible whereas the model presentation in Fig. 4E is smoother and more aesthetically pleasing. As the CPU or GPU must compute more triangles in Fig. 4E than the model Fig. 4C, this smooth structure significantly burdens the system. The model in Fig. 4C was chosen for simulation because it balances computational expense and an aesthetically realistic model.

Force-based soft body simulation

In this particular technique, the soft body’s structure is defined by a triangular mesh, and the constraints exert the necessary pressure to produce the volumetric effect. The forces such as spring, damping, pressure, and offset directional force are considered when modeling each surface point. These forces exert the pressure on a specific area while keeping the total number of vertices constant. As a result of the interaction of these forces, the entire item would shift without altering its original structure.

We define the 3D soft body, SB3D, as

SB3D=S,F,V,N,C

whereas S here defines the soft body, F={Fj||j=0,…,m−1} represents the set of forces for each point j.

V={Vj||j=0,…,m−1} represents the set of velocities for each point j.

N represents the count of total edges.

C represents constraints set being observed for object deformation.

Combined force effect: The composite force, Fj is centered on elasticity control force, pressure control force, damping control force, and a direction offset component from the point j on the body. The composite force Fj is defined by the Eq. (1) as:

(1) Fj=Fsj+Fdj+Fpj+Foj

where Fsj represents the spring force at point j,

Fdj represents the damping force at point j,

Fpj represents the pressure force at point j, and

Foj represents the offset force at point j.

Consider this simulation system with the numerical integration of Hooke’s spring force described through Eq. (2).

(2) Fj=mj.∂2rj→∂t2

The pressure force applied will deform the body shape acting in the direction of normal vector n^ as described in the Eq. (3).

(3) P=P→.n^[Nm2]

where P is a pressure value and n^, is a normal vector to the surface on which pressure force is acting. For calculating specific pressure forces, we multiplied pressure by the surface area, giving us Eq. (4).

(4) FP→=P→.ΔA[N]

Algorithm 1 represents the soft body functioning of this model.

Algorithm 1 Force based soft body algorithm.

Result: Soft body effect	
1. Initialization	
2. Loop over all particles:	
 UpdateVertex(int)	
3. Loop over all faces:	
  (a) UpdateVertexVelocity(vertex velocity)	
  (b) AddPressureForce(3D vertices, force)	
  (c) AddDeformingForce(3D vertices, force)	
  (d) AddForceToVertex(int, 3D vertices, force)	

Detecting user input: A ray is tossed into the scene from the camera location. Unity’s physics engine processes the location data of the object. We move across the mesh surface after pressing the controller trigger buttons to make a cut along the triangles and calculate the locations of the points. An elastic effect results from adding the total forces at the contact locations. The Eq. (5) represents the drag force that each surface point encounters until it reverts to its initial position.

(5) vd=v(1−dΔt)

when the damping force is high, the mesh loses some bounce. Figure 5A presents programmable spring and damping forces variables. The user can easily alter the settings for these variables, giving them full control over how firm the created soft body impression will be.

Figure 5 Mesh deformer input attached to scene camera.

(A) Spring force and damping force variables. (B) Pressure force and offset force variables.

Moving vertices under the pressure force: The AddPressureForce() function iteratively adds the required pressure force to each of the newly discovered locations. To ensure that the vertices are constantly driven into the surface and follow the normal to the contact points, we can add a slight direction offset to this loop. Figure 5B depicts pressure and offset force variables.

Through Eq. (6), we can find the exact distance between vertices and the direction in which force is applied.

(6) Fv=F1+d2

One plus the squared distance ensures the full-force strength even if the distance is zero. Once we know the force, we can use the relation F=ma and a=ΔvΔt to calculate the approximate change in velocity. Considering the mass m = 1 for each vertex, we finally have the Eq. (7).

(7) Δv=FΔt

Now, the vertices can be moved with an absolute velocity updated every frame. The mesh is then given a new shape by adding the moved vertices. The normals for the newly discovered vertices need to be recalculated. Each vertex position is adjusted by the Eq. (8)

(8) Δp=vΔt

Dealing with mesh transformation: An elastic surface is represented by nothing more than a collection of vertices; it lacks a true volume, unlike solid real-world objects. Vertices start to move during cutting as soon as we apply pressure to them with the desired spring and damping effect. Collision and bending constraints in PBD are then used to handle this movement. Since the mesh colliders for the kidney model are unaffected by these forces, the natural shape of the object does not alter throughout the simulation.

Constraint based physics approach

Model points in PBD are connected by the triangular mesh edges and shared connection points. The positions of the points and the force’s direction (velocity is always tangent, constraint force is still normal) are determined directly during the simulation. Because of this, PBD can effectively address the instability issues. Equations (9) and (10) provide the central idea of sequentially resolving the system with constraints of equality and inequality in different time steps.

(9) Cj(xij,...,xinj)=0

(10) Cj(xij,...,xinj)≥0

when simulating, neighboring mesh triangles will respond to bending restrictions under the same stretching influence if one mesh triangle reacts to the stretch amount in the direction normal to the force. The stretching constraint function is described for each connecting edge through Eq. (11).

(11) Cstr(x1,x2)=|x1−x2|−d12

where d12 is the length of the edge at rest position, the bending constraint generated for any two triangles is described through Eq. (12).

(12) Cbend(x1,x2,x3,x4)=arccos(n1.n2)−φ12

where n1 and n2 represent the normal vectors in the rest position and φ12 is the dihedral angle between the triangle 1 and 2. The Eq. (13) for minimizing the total energy defined as:

(13) Etot=∑(kstrCstr2+kbendCbend2)

where kstr and kbend are global stiffness parameters.

This section outlines the fundamental operation of Algorithm 2: the solver iterations compute and project the position constraints. The calculated points are relocated to their projected locations along with updating the point velocities. Distance limitations represent the connective component of the malignant tumor in the kidney. Depending on the stiffness parameter and rest length, each connection is breakable. At a specified break threshold, algorithms tear apart the highlighted locations. However, the deformation does not appear visually plausible because there is just a single layer of a soft structure in our method and no simulation of fascia tissues. The fascia resembles a torn piece of article once the malignant component has been removed from the kidney. The faster solution time brought on by, the smaller number of mesh triangles is still a significant benefit.

Algorithm 2 Position based dynamics.

Result: Constraints handling with collision detection	
1: forall vertices i do GenerateCollisionConstraints (xi→pi)	
2: loop SolverIterations times	
3:    ProjectConstraints(C1,…..,Cn, P1,…..,Pm)	
4: end loop	

Force feedback comparison

The haptic HTC Vive device gives the user a powerful sense of presence. The continuous movement provided by the responsive haptic feedback enhances cognition.

The headset offers a 110-degree field of view, a refresh rate of 90 Hz, and a resolution of 1,080 × 1,200 pixels. According to Table 1, the refresh rate is nearly identical to that of most sensory devices. However, we are enjoying a cost-benefit here with the HTC Vive Pro VR device.

Table 1 Refresh rate comparison of sensory devices (Angelov et al., 2020).

VR devices	Refresh rate	
Oculus Rift S	80 Hz	
HTC Vive Cosmos	90 Hz	
Valve Index	80 Hz	
Samsung HMD Odyssey+	80–144 Hz	
Razer OSVR HDK	90 Hz	
StarVR One	90 Hz	

The SteamVR plugins in Unity allow us to manipulate haptic sensors in communication with organ tissues. Figure 6 demonstrates the haptic device integrated workflow.

Figure 6 HTC Vive haptic device integrated workflow.

The geometry track and the details of the initial interaction are immediately processed. The controller placement and haptic proxy both keep shifting after that. The quick refresh rate gives a credible visual representation of soft tissue deformation in a malignant portion. The PBD-based collision model occasionally removes the triangular meshes to maintain stability. This distortion provides immediate vibrational and visual feedback, which we call the system’s haptic force feedback. The sensory device performance has proved satisfactory while deforming the soft structure in this simulated situation.

System evaluation and feedback

The usability and performance of the system were assessed using the Unity profiler. Since we don’t have to distribute the application on the chosen platform for testing alone, the Unity editor is suitable for quick profiling. The experimentation and performance data from the simulator’s testing on two various machines with various configurations are covered in this section.

Simulator performance comparison using unity profiler

The first simulation experiment was carried out on a system equipped with an Intel(R) Core(TM) i7-7700 CPU running at 3.60 GHz, an x64-based processor, and 8.00 GB of installed memory. The second testing experiment used a high-end graphics system, the GeForce GTX 1060, with Max-Q Design 6.00 GB GDDR5 memory on the card, 1,280 shading units, and 80 texture mapping units. This Max-Q design has the advantage of less power consumption as compared to other graphics cards.

The “CPU/GPU Usage Profiler” in Unity shows information about the duration of various operations and events, including rendering, physics, animations, and scripting. This enumerates every significant region where our system spends most of its time. Figure 7A displays the CPU used for displaying the simulation at a certain timestamp, and Fig. 7B depicts the same simulating system rendering on the GPU. It is clear from Fig. 7B that the majority of the GPU-intensive rendering time is consumed by the “Game Player Loop” and its hierarchy.

Figure 7 Screenshots from unity profiler rendering LPN simulation.

(A) CPU rendering profiler. (B) GPU rendering profiler.

Tables 2 and 3 show the complete component details for Unity’s renderer and memory profilers.

Table 2 CPU and GPU rendering profiler comparison.

Rendering parameters	CPU	GPU	
Total batches	19	9	
Draw calls	19	9	
SetPass calls	20	9	
Triangles	50.2 k	50.2 k	
Vertices	123.9 k	121.9 k	
VRAM usage	9.2 MB	9.1 MB	

Table 3 CPU and GPU memory profiler comparison.

Memory parameters	CPU	GPU	
Total memory	38.9 MB	49.9 MB	
Memory used by unity	29.9 MB	40.2 MB	
Textures	29/299 KB	31/300 KB	
Meshes	11/6.8 MB	6/6.8 MB	
Materials	24/79 KB	26/87 KB	
Game objects in a scene	14	16	

When a significant number of trainers are required, GPU necessitates a high processing unit with a lot of memory, which is fairly expensive and difficult to organize by the hospital authorities. The CPU computation speed was fairly good with a tiny time step, but the user had to make certain sacrifices in terms of deformation performance. The experience was significantly enhanced by the GPU’s capacity to perform parallel processing. With GPU, our simulation’s overall computation time was reduced with a rapid convergence rate.

Face and content validity for simulator performance evaluation

To evaluate the overall performance and usability of the simulator, we have used some face and content validity measures, as presented in Table 4. Each measure has a score range (0–9) (0 indicates a poor score, and nine indicates an excellent score).

Table 4 Face and content validity measures for simulator performance evaluation.

Face validity measures	Content validity measures	
Realistic graphics	Useful for training novice surgeons	
The precision of the platform	Useful for training expert surgeons	
Instrument mapping	Useful for assessing cutting skills	
Anatomy representation	Useful for assessing skill progression	
The interactivity of procedure	Useful for assessing instrument handling	

The nephrology department at Shaikh Zayed Medical Complex Lahore recruited seven final-year students and five interns to participate in this study. The study‘s objectives and the proper use of the simulator were briefly explained to the participants. We are quite appreciative of their assistance in this. Each evaluation metric was given a score according to a scoring system. In Fig. 8, the mean score values from the student and intern groups, representing the face and content validity tests, are plotted.

Figure 8 Simulator performance evaluation.

(A) Mean score from student and intern group for face validity measures (0-Poor, 9-Excellent). (B) Mean score from student and intern group for content validity measures (0-Poor, 9-Excellent).

All participants stated the virtual world was easy to use and that the HTC Vive haptic device helped them practice their skills. Both groups provided favorable feedback about the platform’s precision and interactivity, the metrics with the highest scores. The student group suggested that more realistic graphics be made. The intern group found instrument mapping needed to be more successful when carrying out the process. We treated the laser pointer as the laparoscopic equipment for simulation purposes. The simulator cannot currently teach any experienced surgeons, both groups agreed. It can, however, help new surgeons become more proficient at cutting tissue. Future testing methods and procedures will be increasingly sophisticated to produce more accurate results. Under the direction of medical professionals, we want to improve the anatomy, force feedback, and simulator design and stability.

Game-based system advantages and limitations

Using a constraint-based approach like PBD in combination with Unity has given us considerable advantages, along with some limitations.

Advantages

1) The instability problem is resolved by the constraint-based method. This technique provides control and assistance for physical-based effects.

2) A time-step approach updates the point’s locations and velocities, reducing the processing time.

3) With certain features sacrificed, game-based simulators are less expensive and aid in quick prototyping.

4) The user-friendly environment offered by contemporary gaming engines like Unity makes it possible to simulate medical procedures, collaborative tasks, and emergencies.

Limitations

1) For a more complicated system (multilayer tissue models), the total calculation time for simulation increases.

2) A game engine’s main purpose is to create games. The human anatomy is too intricate to reproduce by a physics-based game engine accurately.

HTC Vive as a surgical trainer

Numerous studies have demonstrated that virtual simulators do not offer realistic force or haptic feedback, nor are they extremely accurate and constant. Another big disadvantage of these high-fidelity VR simulators is their exorbitant price. Additionally, the annual cost of maintaining hardware and software is substantially higher. Even though these gadgets significantly impact surgical training as a whole, their high cost can make them an undesirable option compared to alternative resources that are similarly priced. An affordable option that offers a vibrating effect for tactile movement is the HTC Vive head-mounted display. This effect has been considered haptic feedback from the system. We have contrasted our suggested system’s cost and functionality with some of the more common virtual simulators offered commercially. A cost comparison analysis is displayed in Table 5. Our research has demonstrated that, with some performance and feedback functionality compromises, the HTC Vive can be the most reasonably priced VR-based surgical trainer when used with an open-source game engine.

Table 5 Comparative cost analysis of various virtual simulators.

Device	Description/Characteristics	Cost	
MIST VR	Advance technology based on surgical practices curriculum. An expanding library of modules. Used in suturing, knot-tying, needle passing and stitching	$16000–$25000	
LAPSIM Essence	Camera and instrument navigation, coordination, lifting, grasping, and fine dissection	$5900	
LAP Mentor III	Portable, cost-effective, height adjustable tower, non-haptic, ideal for team training	$500	
Oculus Rift S	Improved optics with vivid colors, Peed and comfort, intuitive and, realistic precision	$770	
Samsung Odyssey	Inside-out tracking, high resolution, high-end integrated audio, and wider field of view	$500	
HTC Vive	Fully immersive, intuitive controls and gestures, realistic haptic feedback, and eye relief adjustments	$399	

Conclusion and future improvements

This research must be seen as a first step toward developing a functional LPN training simulator. This article demonstrates the effective deformation of a soft body of simplex meshes at minimal computing cost. A practical and affordable solution for surgical training is a virtual, realistic LPN simulator built on Unity3D. The medical users gave the tests positive feedback, demonstrating more control over the elasticity and force used during the laparoscopic surgery and improving their training experience. The use of HTC Vive in LPN training, which has not yet been investigated in this field, is another innovative aspect of this work.

We propose the following future improvements: 1) Additional needs such as force feedback constraints must be established to enable a more realistic behavior.

2) For improved anatomical depiction, the organ and associated anatomical features (adhesive restrictions, fascia tissues, and ureter) can be mapped from a huge quantity of data utilizing image analysis and segmentation.

3) The competency goals of the simulator can be established for evaluation reasons, and simulated process proficiency scores can be computed.

Additional Information and Declarations

Competing Interests

Author Contributions

Data Availability

Muhammad Asif is an Academic Editor for PeerJ Computer Science.

Fareeha Rasheed conceived and designed the experiments, performed the experiments, analyzed the data, performed the computation work, authored or reviewed drafts of the article, and approved the final draft.

Faisal Bukhari conceived and designed the experiments, performed the experiments, analyzed the data, performed the computation work, prepared figures and/or tables, authored or reviewed drafts of the article, and approved the final draft.

Waheed Iqbal conceived and designed the experiments, analyzed the data, prepared figures and/or tables, authored or reviewed drafts of the article, and approved the final draft.

Muhammad Asif conceived and designed the experiments, analyzed the data, prepared figures and/or tables, authored or reviewed drafts of the article, and approved the final draft.

Hafiza Ayesha Hoor Chaudhry conceived and designed the experiments, authored or reviewed drafts of the article, and approved the final draft.

The following information was supplied regarding data availability:

The code and the dataset generated by our own simulator, Unity 3D are available at Zenodo: Fareeha. (2023). A Low-cost Unity-based Virtual Training Simulator for Laparoscopic Partial Nephrectomy using HTC Vive. https://doi.org/10.5281/zenodo.8054034.

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
