# Peer review of "A low-cost unity-based virtual training simulator for laparoscopic partial nephrectomy using HTC Vive"

_PeerJ Computer Science, doi:10.7717/peerj-cs.1627_

## Round 0.1 · original submission · Major Revisions

The authors must respond to reviewers suggestions and correct where it is necessary!

**Language Note:** The review process has identified that the English language must be improved. PeerJ can provide language editing services - please contact us at copyediting@peerj.com for pricing (be sure to provide your manuscript number and title). Alternatively, you should make your own arrangements to improve the language quality and provide details in your response letter. – PeerJ Staff

Reviewer 1 ·

Basic reporting

The paper introduces an affordable system for simulating Laparoscopic Partial Nephrectomy, utilizing HTC Vive technology. Such cost-effective systems hold great potential in enhancing the adoption of simulation tools among students. In general, the paper is excellently written and structured, and the state-of-the-art section offers a comprehensive literature review. The authors demonstrate a profound understanding of the surgical environment and the challenges it poses. Additionally, the majority of the prepared figures (excluding screenshots) effectively illustrate the overall schema. The emphasis on conducting research that caters to various budget constraints for training scenarios is commendable. As a result, the reviewer wholeheartedly supports the authors for their remarkable work and valuable contributions to training scenarios. The paper should be proofread for some English errors.
The reviewer recommends proofreading the document before publication. Some of the minor errors encountered are:
"C represents a constraints set" -> "C represents a constraints set" (remove the extra "a")
"less computing time, but high solving speed" is indeed redundant since "less computing time" and "high solving speed" convey the same idea. It would be more appropriate to rephrase it to avoid repetition.

Experimental design

No Comment

Validity of the findings

No Comment

Additional comments

No Comment

Reviewer 2 ·

Basic reporting

This work describes a surgical simulator for initial Laparoscopic Partial Nephrectomy (LPN) training. This study used an open-source game engine and a low-cost virtual reality (VR) device with visual and haptic input to mimic the simulator. Such cost-effective systems hold great potential to enhance the adoption of simulation tools among students.
The 3D virtual kidney modelling aspects have been thoroughly detailed. The comprehensiveness with which the force-based soft body simulation method is discussed is commendable. Furthermore, the majority of the figures provided adequately depict the general schema. The authors also discussed using survey-based questionnaires for face and content validation. The paper is well-written and structured, and the cutting-edge portion provides a thorough literature review.
The paper also covers computing efficiency and interoperability with CPU and GPU-based devices. The paper concludes that the UNITY3D training simulator with soft bodies made of simplex meshes may be deformed efficiently and at a minimal computational cost. After carefully weighing the benefits and downsides of the technology, the paper suggests that HTC Vive is a viable option for LPT training.
Some of the errors encountered are:
1. On page 4, section 4.2, "C represents constraints set" should be replaced with "C represents constraints set."

2. On page 3, section 3, step 3, "With the help of two laser emitting controllers, trainees can easily mark and precisely cut along the cancerous structure, avoiding damage to the nearest tissues" should be replaced with "avoiding damage to the nearby tissues."

Experimental design

The authors have crafted an impressive virtual training simulator for laparoscopic partial nephrectomy, but to truly stand out, they could spice things up by adding more comparative details with other simulators in the field. This would help showcase the distinct advantages and unique features of their work.

Validity of the findings

The paper presents an exciting low-cost surgical Unity-based Virtual Training Simulator for laparoscopic partial nephrectomy using the HTC Vive. The authors highlight several noteworthy contributions. However, it would be wonderful to see more details in the results and comparative analysis section. A deeper dive into the experimental setup, user studies, and specific metrics used to evaluate the simulator's performance would add more weight to their claims.

Additional comments

Carefully proofread the grammar, syntax, and clarity to ensure the paper's overall readability.

---

## Round 0.2 · accepted · Accept

The paper was well improved according to reviewers' comments.

Reviewer 1 ·

Basic reporting

as in comments section

Experimental design

as in comments section

Validity of the findings

as in comments section

Additional comments

The manuscript is improved and all my comments are incorporated. I have no further concerns and recommend it in the current form as accepted.

Reviewer 2 ·

Basic reporting

The authors have effectively incorporated all the feedback. I concur with moving forward to accepting the article in its present form.

Experimental design

The authors have incorporated the suggestions, accordingly.

Validity of the findings

The results section has also been revised as requested.

---

## Author Rebuttal · Round 0.2

# Response to Referee Report

Manuscript # ———-, "A Low-cost Unity-based Virtual Training Simulator for Laparoscopic Partial Nephrectomy using HTC Vive"

August 28, 2023

We would like to thank the respected Editor-in-Chief, Associate Editor, and reviewers for their comments, which helped us improve the quality of our paper. We have incorporated all the suggestions and revised the manuscript accordingly. In this document, we explain our response to each reviewer's comment.

## Reviewer #1

The paper introduces an affordable system for simulating Laparoscopic Partial Nephrectomy, utilizing HTC Vive technology. Such cost-effective systems hold great potential in enhancing the adoption of simulation tools among students. In general, the paper is excellently written and structured, and the state-of-the-art section offers a comprehensive literature review. The authors demonstrate a profound understanding of the surgical environment and the challenges it poses. Additionally, the majority of the prepared figures (excluding screenshots) effectively illustrate the overall schema. The emphasis on conducting research that caters to various budget constraints for training scenarios is commendable. As a result, the reviewer wholeheartedly supports the authors for their remarkable work and valuable contributions to training scenarios. Some of the minor errors encountered are:

**Comment 1:** *"C represents a constraints set" with "C represents a constraints set" (remove the extra "a"))*

> **Author Response:** The modified version of the amended manuscript reflects our agreement with the esteemed reviewer.
> **Author Action:** "C represents a constraints set" has been corrected by changing "C represents constraints set." on Page 6, Line 204 in the revised manuscript.

**Comment 2:** *"less computing time, but high solving speed" is indeed redundant since "less computing time" and "high solving speed" convey the same idea. It would be more appropriate to rephrase it to avoid repetition.*

> **Author Response:** We agree with the respected reviewer, and the revised manuscript has been updated accordingly.
> **Author Action:** Page 2, Line 75 has been changed to read, "Our suggested training method based on Unity follows a more simple deformation representation with less processing time as compared to existing high fidelity pricey trainers." in order to make the appropriate changes in the revised manuscript.
>
> Page 14, Line 356 has been changed to read, "In this paper, we have demonstrated the effective deformation of a soft body of simplex meshes at minimal computing cost." in order to make the appropriate changes in the revised manuscript.

**Comment 3:** *The paper should be proofread for some English errors. The reviewer recommends proofreading the document before publication.*

> **Author Response:** We thank the reviewer for the feedback and suggestion.
> **Author Action:** The revised manuscript has been proofread for language and grammatical errors.

## Reviewer #2

This work describes a surgical simulator for initial Laparoscopic Partial Nephrectomy (LPN) training. This study used an open-source game engine and a low-cost virtual reality (VR) device with visual and haptic input to mimic the simulator. Such cost-effective systems hold great potential to enhance the adoption of simulation tools among students. The 3D virtual kidney modeling aspects have been thoroughly detailed. The comprehensiveness with which the force-based soft body simulation method is discussed is commendable. Furthermore, the majority of the figures provided adequately depict the general schema.

The authors also discussed using survey-based questionnaires for face and content validation. The paper is well-written and structured, and the cutting-edge portion provides a thorough literature review. The paper also covers computing efficiency and interoperability with CPU and GPU-based devices. The paper concludes that the UNITY3D training simulator with soft bodies made of simplex meshes may be deformed efficiently and at a minimal computational cost. After carefully weighing the benefits and downsides of the technology, the paper suggests that HTC Vive is a viable option for LPT training.
Some of the errors encountered are:

*Comment 1: "C represents constraints set" should be replaced with "C represents constraints set."*

**Author Response:** The modified version of the amended manuscript reflects our agreement with the esteemed reviewer.
**Author Action:** "C represents a constraints set" has been corrected by changing "C represents constraints set." on Page 6, Line 204 in the revised manuscript.

*Comment 2: "With the help of two laser emitting controllers, trainees can easily mark and precisely cut along the cancerous structure, avoiding damage to the nearest tissues" should be replaced with "avoiding damage to the nearby tissues."*

**Author Response:** We agree with the respected reviewer, and the revised manuscript has been updated accordingly.
**Author Action:** On Page 4 and Line 166, the essential modification has been made by changing "Trainees can quickly mark and precisely cut along the malignant structure with the aid of two laser emitting controllers while minimizing harm to the nearby tissues." in the revised manuscript.

*Comment 3: The authors have crafted an impressive virtual training simulator for laparoscopic partial nephrectomy, but to truly stand out, they could spice things up by adding more comparative details with other simulators in the field. This would help showcase the distinct advantages and unique features of their work.*

**Author Response:** We agree with the esteemed reviewer that this type of comparison has a significant influence.
**Author Action:** On Page 9, Table 1 prvides the HTC Vive's force feedback comparison with that of other haptic devices. On Page 13, Table 5, a cost comparison of several devices is presented. The significant benefits of using the HTC Vive haptic device are amply demonstrated by both kinds of comparisons.

*Comment 4: The paper presents an exciting low-cost surgical Unity-based Virtual Training Simulator for laparoscopic partial nephrectomy using the HTC Vive. The authors highlight several noteworthy contributions. However, it would be wonderful to see more details in the results and comparative analysis section. A deeper dive into the experimental setup, user studies, and specific metrics used to evaluate the simulator's performance would add more weight to their claims.*

**Author Response:** We appreciate the reviewer's comments and recommendations. The report already includes force feedback and cost comparative analysis. We concur with the respected reviewer that improving the experimental environment and developing specific measures for evaluating the simulator's efficacy would support our claims, but doing so is now outside the purview of our experimental effort. This kind of setup and results will undoubtedly be included in our upcoming work.

*Comment 5: Carefully proofread the grammar, syntax, and clarity to ensure the paper's overall readability.*

**Author Response:** We thank the reviewer for the feedback and suggestion.
**Author Action:** The revised manuscript has been proofread for language and grammatical errors.